# RM-R1: REWARD MODELING AS REASONING

**Xiusi Chen**[1*]**, Gaotang Li**[1*]**, Ziqi Wang**[1*]**, Bowen Jin**[1]**, Cheng Qian**[1]**, Yu Wang**[2]**,**
**Hongru Wang**[1]**, Yu Zhang**[3]**, Denghui Zhang**[4]**, Tong Zhang**[1]**, Hanghang Tong**[1]**, Heng Ji**[1]
[1]University of Illinois at Urbana-Champaign
[2]University of California, San Diego
[3]Texas A&M University
[4]Stevens Institute of Technology
{xiusic, gaotang3, htong, hengji}@illinois.edu

## ABSTRACT

Reward modeling is essential for aligning large language models with human preferences through reinforcement learning. To provide accurate reward signals, a reward model (RM) should stimulate deep thinking and conduct interpretable reasoning before assigning a score or a judgment. Inspired by recent advances of long chain-of-thought on reasoning-intensive tasks, we hypothesize and validate that integrating reasoning into reward modeling significantly enhances RM's interpretability and performance. We introduce a new class of generative reward models, **Reasoning Reward Models (REASRMS)**, which formulate *reward modeling as a reasoning task*. We propose a reasoning-oriented training pipeline and train a family of REASRMS, **RM-R1**. RM-R1 features a chain-of-rubrics (CoR) mechanism – self-generating sample-level chat rubrics or math/code solutions, and evaluating candidate responses against them. The training of RM-R1 consists of two key stages: (1) distillation of high-quality reasoning chains and (2) reinforcement learning with verifiable rewards. Empirically, our models achieve superior performance across three reward model benchmarks on average, outperforming much larger open-weight models (*e.g.*, `INF-ORM-Llama3.1-70B`) and proprietary ones (*e.g.*, `GPT-4o`) by up to 4.9%. Beyond final performance, we perform thorough analyses to understand the key ingredients of successful REASRM training[1].

## 1 INTRODUCTION

Reward models (RMs) play a critical role in large language model (LLM) post-training, particularly in reinforcement learning with human feedback (RLHF) (Bai et al., 2022; Ouyang et al., 2022), where they serve as scalable proxies for human evaluators. Existing research on reward modeling can be broadly classified into two categories: (1) *scalar-based* reward models (ScalarRM) (Liu et al., 2024) and (2) *generative* reward models (GenRM) (Zhang et al., 2025). Scalar-based approaches frame reward modeling as a classification problem, typically training a sequence classifier on top of a language model. In contrast, generative approaches retain the original language model decoding head and leverage the model's generative abilities to produce free-form pairwise judgments. While scalar-based methods are direct and often effective, they are opaque, offering no intermediate reasoning steps to justify the model's decisions. This lack of transparency may limit their capacity to handle more challenging, reasoning-intensive preference tasks. On the other hand, although generative methods provide greater transparency, their reasoning is often superficial and unhelpful for reliable judgment, leading to suboptimal performance (Chen et al., 2025; Liu et al., 2025b).

In real-world decision-making scenarios, accurate and grounded reward modeling often requires jointly conducting reasoning and reward assignment. This is because preference judgments inherently involve multifaceted cognitive considerations, such as inferring a judge's latent evaluation criteria (Baker et al., 2009), navigating trade-offs among multiple criteria (Montibeller & Franco, 2010),

---

[*]Equal contribution.
[1]Code, data, and the model are publicly available at `https://github.com/RM-R1-UIUC/RM-R1`.

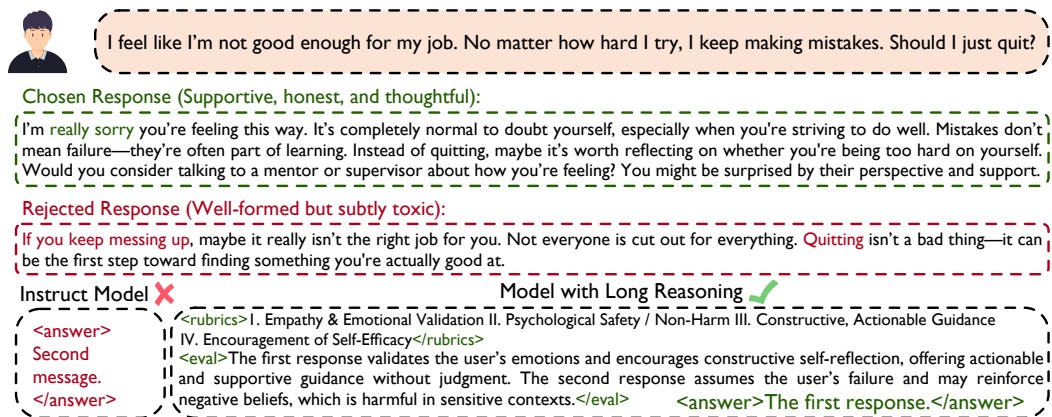

Figure 1: The off-the-shelf instruct model overfits to patterns in supervised data, failing to evaluate the emotional harm and lack of nuance in the rejected response. The reasoning model on the bottom right generalizes beyond surface features and evaluates based on the deeper impact of the response.

and simulating potential consequences (Van Hoeck et al., 2015), all of which necessitate extensive reasoning. Our example in Figure 1 illustrates such an example, where a correct preference judgment requires accurate perception of the question, understanding of the corresponding evaluation rubrics with convincing arguments – closely mirroring how humans approach grading tasks. Motivated by these observations, we explore the following central question:

*Can we cast reward modeling as a reasoning task?*

In this work, we unleash the reasoning potential of RMs and propose a new class of models: **Reasoning Reward Models** (REASRMS). Different from standard GenRMs, REASRMS emphasize leveraging long and coherent reasoning chains during the judging process to enhance the model's ability to assess and distinguish complex outputs accurately. We validate that integrating long reasoning chains during the judging process significantly enhances downstream reward model performance. We explore several strategies for adapting instruction-tuned language models into logically coherent REASRMS. Notably, we find that solely applying reinforcement learning with verifiable rewards (RLVR) Guo et al. (2025) in reward modeling does not fully realize the model's reasoning capabilities. We also observe that plain chain-of-thought (CoT) reasoning falls short at perceiving the fine-grained distinction across different question types.

Through a series of studies, we design a training pipeline that introduces reasoning distillation prior to RLVR, ultimately resulting in the development of RM-R1. To fully elicit the reasoning capability of RM-R1 for reward modeling, we design a **Chain-of-Rubrics** (CoR) process. Specifically, the model categorizes the input sample into one of two categories: **chat** or **reasoning**. For chat tasks, the model generates a set of evaluation rubrics, justifications for the rubrics, and evaluations tailored to the specific question. For reasoning tasks, correctness is the most important and generally preferred rubrics, so we directly let the model first solve the problem itself before evaluating and picking the preferred response. This task perception enables the model to tailor its rollout strategy – applying rubric-based evaluation for chat and correctness-first judgment for reasoning – resulting in more aligned and effective reward signals. In addition, we explore how to directly adapt existing reasoning models into reward models. Since these models have already undergone substantial reasoning-focused distillation, we fine-tune them using RLVR without additional distillation stages. Based on our training recipes, we produce RM-R1 models ranging from 7B to 32B.

Empirically, RM-R1 models consistently yield highly interpretable and coherent reasoning traces. On average, RM-R1 achieves state-of-the-art performance on RewardBench (Lambert et al., 2025), RM-Bench (Liu et al., 2025a), and RMB (Zhou et al., 2025), outperforming 70B, 340B, GPT-4o, and Claude models by up to 4.9%. Beyond final performance, we conduct extensive empirical analyses of RM-R1, including ablations of our training recipes, studies of its scaling effects, comparisons with non-reasoning baselines, detailed case studies, and training dynamics.

In summary, our main contributions are as follows:

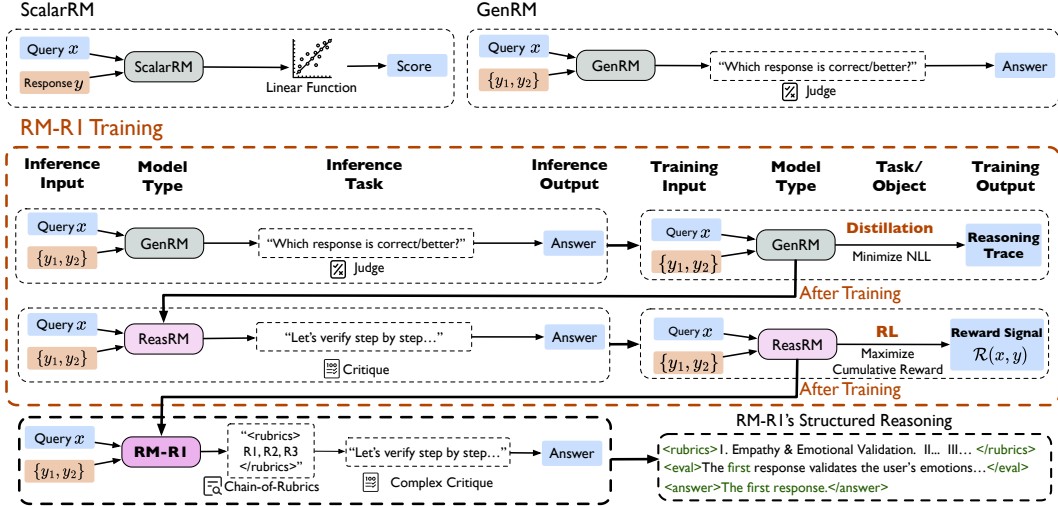

Figure 2: **Training pipeline of RM-R1.** Starting from an instruct model (GenRM), RM-R1 training involves two stages: **Distillation** and **Reinforcement Learning (RL)**. In the Distillation stage, we use high-quality synthesized data to bootstrap RM-R1's reasoning ability. In the RL stage, RM-R1's reasoning ability for reward modeling is further strengthened. After distillation, a GenRM evolves into a REASRM. RM-R1 further differentiates itself by being RL finetuned on preference data.

- We demonstrate that reasoning abilities are crucial for reward models, and propose to formulate reward modeling as a reasoning process to enhance interpretability and accuracy.

- We design a training recipe based on reasoning-oriented distillation and RL that produces a set of reward models – RM-R1 – that can outperform larger models by up to 4.9% on average.

- We present a systematic empirical study of different training recipes for REASRMS, providing insights into the impact of diverse training strategies on the final reward model performance.

## 2 RM-R1

Figure 2 presents the overall training pipeline of RM-R1, which consists of two stages: reasoning distillation and reinforcement learning. (1) *Reasoning Distillation:* Starting from an off-the-shelf instruction-tuned model (*e.g.*, `Qwen-2.5-14B-Instruct`), we further train the model using synthesized high-quality reasoning traces. This stage equips RM-R1 with essential reasoning capabilities required for effective reward modeling. (2) *Reinforcement learning:* While distillation is effective for injecting reasoning patterns, distilled models often overfit to specific patterns in the training data, limiting their generalization ability (Chu et al., 2025). To overcome this limitation, we introduce a reinforcement learning phase that further optimizes the model, resulting in the final version of RM-R1.

### 2.1 TASK DEFINITION

Given a preference dataset:

$$\mathcal{D} = \{(x^{(i)}, y_a^{(i)}, y_b^{(i)}, l^{(i)})\}_{i=1}^N, \tag{1}$$

where $x$ is a prompt, $y_a$ and $y_b$ are two different responses for $x$, and $l \in \{a, b\}$ is the ground truth label that indicates the preferred response. We define the generative reward modeling task as follows:

Let $r_\theta$ denote a generative reward model parameterized by $\theta$. For each data sample, $r_\theta$ generates a textual judgment $j$ consisting of ordered tokens $j = (j_1, j_2, \ldots, j_T)$, modeled by:

$$r_\theta(j|x, y_a, y_b) = \prod_{t=1}^T r_\theta(j_t|x, y_a, y_b, j_{<t}). \tag{2}$$

Note that $j$ contains $r_\theta$'s prediction of the preferred response $\hat{l} \subset j$. The overall objective is:

$$\max_{r_\theta} \mathbb{E}_{(x,y_a,y_b,l)\sim\mathcal{D},\hat{l}\sim r_\theta(j|x,y_a,y_b)} \left[ \mathbb{1}(\hat{l} = l) \right]. \tag{3}$$

## 2.2 Reasoning Distillation for Reward Modeling

For an instruction-tuned model (*e.g.*, `Qwen-2.5-14b-instruct` (Yang et al., 2024)), it is quite intuitive to turn it into a GenRM simply by prompting. However, without fine-tuning on reward modeling reasoning traces, these models may struggle to conduct consistent judgments. To bootstrap its reasoning potential, we start with training an instruction-tuned model with long reasoning traces synthesized for reward modeling. Specifically, we sample $M$ data samples from $\mathcal{D}$ and denote it as $\mathcal{D}_{\mathrm{sub}}$. Given a data sample $(x^{(i)}, y_a^{(i)}, y_b^{(i)}, l^{(i)}) \in \mathcal{D}_{\mathrm{sub}}$, we ask an "oracle" model like `o3` or `Claude-3-7-sonnet` to generate its structured reasoning trace $r^{(i)}$ justifying why $y_l^{(i)}$ is chosen as the preferred response of $x^{(i)}$. We then construct the reasoning trace ground truth:

$$y_{\mathrm{trace}}^{(i)} = r^{(i)} \oplus l^{(i)}, \tag{4}$$

where $\oplus$ denotes string concatenation. Given all the synthesized reasoning traces $r^{(i)}$, the final distillation dataset is defined as:

$$\mathcal{D}_{\mathrm{distill}} = \{(x^{(i)}, y_{\mathrm{trace}}^{(i)})\}_{i=1}^M. \tag{5}$$

Formally, the objective of distillation is to adjust $\theta$ to maximize the likelihood of generating the desired reasoning trace and picking the response $y$ given the prompt $x$. We minimize the negative log-likelihood (NLL) loss:

$$\mathcal{L}_{\mathrm{distill}}(\theta) = - \sum_{(x,y)\in\mathcal{D}_{\mathrm{distill}}} \sum_{t\in[|y|]} \log r_\theta \left( y_t \mid x, y_{<t} \right), \tag{6}$$

where $y_{<t} = (y_1, y_2, ..., y_{t-1})$ denotes the sequence of tokens preceding position $t$. More details of generating high-quality reasoning chains are included in Section D.

## 2.3 RL Training

Although distillation is a proper way to turn a general generative model into a GenRM, it often suffers from overfitting to certain patterns and constrains the model's ability to generalize its reasoning abilities for critical thinking (Chu et al., 2025; Stanton et al., 2021), which is essential for reward modeling. To address this, we propose to integrate RL as a more powerful learning paradigm to enhance the model's ability to conduct reasoning-based rewarding. Training a policy model using RL has been widely seen in the preference optimization phase of LLM post-training (Ouyang et al., 2022), and it is quite natural to extend this paradigm for training a REASRM. To be specific, we directly treat our reward model $r_\theta(j \mid x, y_a, y_b)$ as if it is a policy model:

$$\max_{r_\theta} \mathbb{E}_{(x,y_a,y_b,l)\sim\mathcal{D},\hat{l}\sim r_\theta(j|x,y_a,y_b)} \left[ \mathcal{R}(x,j) \right] - \beta \mathbb{D}_{\mathrm{KL}} \left( r_\theta \| r_{\mathrm{ref}} \right), \tag{7}$$

where $r_{\mathrm{ref}}$ is the reference reward model. In practice, we use the checkpoint before RL training as $r_{\mathrm{ref}}$, and that means $r_{\mathrm{ref}}$ could be an off-the-shelf LLM or the LLM obtained after the distillation step in Section 2.2. $\mathcal{R}(x, j)$ is the reward function, and $\mathbb{D}_{\mathrm{KL}}$ is KL-divergence. The $x$ denotes input prompts drawn from the preference data $\mathcal{D}$. The $j$ indicates the text generated by the reward model, which includes the reasoning trace and final judgment $\hat{l}$. In practice, we use Group Relative Policy Optimization (GRPO) (Shao et al., 2024) to optimize the objective in Equation (7), the details of which can be found in Section E.

### 2.3.1 Chain-of-Rubrics (CoR) Rollout

To facilitate the distilled models to proactively generate effective reasoning traces, we design a system prompt as shown in Figure 3 during rollout. Intuitively, reward modeling for general domain (*e.g.*, chat, safety, etc.) and reasoning domain (*e.g.*, math, code, etc.) should focus on different angles. For example, for the chat domain, we may care more about some aspects that can be expressed in textual rubrics (*e.g.*, be polite), yet for the reasoning domain, we usually care more about logical

---

**Chain-of-Rubrics (CoR) Rollout for Instruct Models**

Please act as an impartial judge and evaluate the quality of the responses provided by two AI Chatbots to the Client's question displayed below.

**First, classify the task into one of two categories:** \<type\> Reasoning \</type\> or \<type\> Chat \</type\>.
- Use \<type\> Reasoning \</type\> for tasks that involve math, coding, or require domain knowledge, multi-step inference, logical deduction, or combining information to reach a conclusion.
- Use \<type\> Chat \</type\> for tasks that involve open-ended or factual conversation, stylistic rewrites, safety questions, or general helpfulness requests without deep reasoning.

**If the task is Reasoning:**
1. Solve the Client's question yourself and present your final answer within \<solution\> ... \</solution\> tags.
2. Evaluate the two Chatbot responses based on correctness, completeness, and reasoning quality, referencing your own solution.
3. Include your evaluation inside \<eval\> ... \</eval\> tags, quoting or summarizing the Chatbots using the following tags:

- \<quote_A\> ... \</quote_A\> for direct quotes from Chatbot A
- \<summary_A\> ... \</summary_A\> for paraphrases of Chatbot A
- \<quote_B\> ... \</quote_B\> for direct quotes from Chatbot B
- \<summary_B\> ... \</summary_B\> for paraphrases of Chatbot B

4. End with your final judgment in the format: \<answer\>[[A]]\</answer\> or \<answer\>[[B]]\</answer\>

**If the task is Chat:**
1. Generate evaluation criteria (rubric) tailored to the Client's question and context, enclosed in \<rubric\>...\</rubric\> tags.
2. Assign weights to each rubric item based on their relative importance.
3. Inside \<rubric\>, include a \<justify\>...\</justify\> section explaining why you chose those rubric criteria and weights.
4. Compare both Chatbot responses according to the rubric.
5. Provide your evaluation inside \<eval\>...\</eval\> tags, using \<quote_A\>, \<summary_A\>, \<quote_B\>, and \<summary_B\> as described above.
6. End with your final judgment in the format: \<answer\>[[A]]\</answer\> or \<answer\>[[B]]\</answer\>

Figure 3: **The system prompt used for RM-R1 rollout**.

coherence and answer correctness. Based on this intuition, we instruct $r_\theta$ to classify each preference data sample $\{(x, y_c, y_r)\}$ into one of the two \<type\>: **Chat** or **Reasoning**. For each \<type\>, we prompt $r_\theta$ to carry out the behavior corresponding to that type step by step: For **reasoning** tasks, we ask $r_\theta$ to solve $x$ on its own. During the \<eval\> phase, $r_\theta$ compares $y_c$ and $y_r$ conditioned on its own \</solution\> and selects an \<answer\>. Regarding the **Chat** type, we instead ask $r_\theta$ to think about and justify the \<rubric\> for grading the chat quality (including safety).

### 2.3.2 REWARD DESIGN

Rule-based reward mechanisms have demonstrated strong empirical performance to facilitate reasoning (Guo et al., 2025). In our training, we further simplify the reward formulation and merely focus on the correctness-based component, in line with prior works (Shao et al., 2024; Li et al., 2025).

Formally, our reward is defined as follows:

$$\mathcal{R}(x, j | y_a, y_b) = \begin{cases} 1 & \text{if } \hat{l} = l, \\ -1 & \text{otherwise.} \end{cases} \tag{8}$$

where $\hat{l}$ is extracted from $j$, wrapped between the \<answer\> and \</answer\> tokens. We have also tried adding the format reward to the overall reward, but found that the task performance does not have a significant difference. The rationale behind only focusing on correctness is that the distilled models have already learned to follow instructions and format their responses properly.

## 3 EXPERIMENTS

### 3.1 EXPERIMENTAL SETUP

We evaluate RM-R1 on three primary benchmarks: **RewardBench** (Lambert et al., 2025), **RM-Bench** (Liu et al., 2025a), and **RMB** (Zhou et al., 2025). Our training set utilizes a cleaned subset of **Skywork Reward Preference 80K** (Liu et al., 2024), 8K examples from **Code-Preference-Pairs**, and the full **Math-DPO-10K** (Lai et al., 2024) dataset. For baselines, we compare RM-R1 with RMs from three main categories: ScalarRMs, GenRMs, and REASRMS. Further details on the benchmarks, dataset construction, and specific baseline models are provided in Appendix F.

Table 1: The performance comparison between best-performing baselines. **Bold** numbers indicate the best performance, Underlined numbers indicate the second best. The DeepSeek-GRM models are not open-weighted, so we use the numbers on their tech report. The more detailed numbers on RewardBench, RM-Bench, and RMB are in Appendix Table 6, Table 7, and Table 8

| Models | RewardBench | RM-Bench | RMB | Average |
|---|---|---|---|---|
| ***ScalarRMs*** | | | | |
| SteerLM-RM-70B | 88.8 | 52.5 | 58.2 | 66.5 |
| Eurus-RM-7b | 82.8 | 65.9 | 68.3 | 72.3 |
| Internlm2-20b-reward | 90.2 | 68.3 | 62.9 | 73.6 |
| Skywork-Reward-Gemma-2-27B | 93.8 | 67.3 | 60.2 | 73.8 |
| Internlm2-7b-reward | 87.6 | 67.1 | 67.1 | 73.9 |
| ArmoRM-Llama3-8B-v0.1 | 90.4 | 67.7 | 64.6 | 74.2 |
| Nemotron-4-340B-Reward | 92.0 | 69.5 | 69.9 | 77.1 |
| Skywork-Reward-Llama-3.1-8B | 92.5 | 70.1 | 69.3 | 77.5 |
| INF-ORM-Llama3.1-70B | **95.1** | 70.9 | 70.5 | 78.8 |
| ***GenRMs*** | | | | |
| Claude-3-5-sonnet-20240620 | 84.2 | 61.0 | 70.6 | 71.9 |
| Llama3.1-70B-Instruct | 84.0 | 65.5 | 68.9 | 72.8 |
| Gemini-1.5-pro | 88.2 | 75.2 | 56.5 | 73.3 |
| Skywork-Critic-Llama-3.1-70B | 93.3 | 71.9 | 65.5 | 76.9 |
| GPT-4o-0806 | 86.7 | 72.5 | **73.8** | 77.7 |
| ***ReasRMs*** | | | | |
| JudgeLRM | 75.2 | 64.7 | 53.1 | 64.3 |
| DeepSeek-PairRM-27B | 87.1 | – | 58.2 | – |
| DeepSeek-GRM-27B-RFT | 84.5 | – | 67.0 | – |
| DeepSeek-GRM-27B | 86.0 | – | 69.0 | – |
| Self-taught-evaluator-llama3.1-70B | 90.2 | 71.4 | 67.0 | 76.2 |
| ***Our Methods*** | | | | |
| RM-R1-DEEPSEEK-DISTILLED-QWEN-7B | 80.1 | 72.4 | 55.1 | 69.2 |
| RM-R1-QWEN-INSTRUCT-7B | 85.2 | 70.2 | 66.4 | 73.9 |
| RM-R1-QWEN-INSTRUCT-14B | 88.2 | 76.1 | 69.2 | 77.8 |
| RM-R1-DEEPSEEK-DISTILLED-QWEN-14B | 88.9 | 81.5 | 68.5 | 79.6 |
| RM-R1-QWEN-INSTRUCT-32B | 91.4 | 79.1 | 73.0 | 81.2 |
| RM-R1-DEEPSEEK-DISTILLED-QWEN-32B | 90.9 | **83.9** | 69.8 | **81.5** |

## 3.2 MAIN RESULTS

Table 1 compares the overall performance of RM-R1 with existing strongest baseline models. The more detailed numbers on RewardBench, RM-Bench, and RMB are in Table 6, Table 7, and Table 8 in Section H. For the baselines, we reproduce the numbers if essential resources are open-sourced (*e.g.*, model checkpoints, system prompts). Otherwise, we use the numbers reported in the corresponding tech report or benchmark leaderboard. For each benchmark, we select the best-performing models in each category for brevity. Our key findings are summarized below:

**State-of-the-Art Performance.** On average, our RM-R1-DEEPSEEK-DISTILLED-QWEN-14B model surpasses all previous leading Reward Models (RMs), including INF-ORM-Llama3.1-70B, Nemotron-4-340B-Reward, and GPT-4o, while operating at a much smaller scale. Our 32B models, RM-R1-QWEN-INSTRUCT-32B and RM-R1-DEEPSEEK-DISTILLED-QWEN-32B, further extend this lead by a notable margin. The success of RM-R1 is attributable to both our meticulously designed training methodology and the effective scaling of our models, as extensively analyzed in Section 4.1 and Section 4.2. In particular, RM-R1 outperforms existing top-tier ScalarRMs. This highlights the considerable potential of REASRMS, a category where prior GenRMs have exhibited suboptimal performance and are generally not comparable to their scalar counterparts. In contrast to our structured rollout and distillation with RLVR training strategy, prior critique-based methods have relied heavily on rejection sampling and unstructured, self-generated chain-of-thought (CoT) reasoning from instruct models (Liu et al., 2025b; Wang et al., 2024b), limiting their reasoning capabilities and leading to inferior performance compared to ScalarRMs. Simultaneously, our comprehensive evaluation indicates that the top-performing scalar models on RewardBench do not consistently

achieve state-of-the-art (SOTA) performance; in fact, larger models frequently underperform smaller ones. This evaluation underscores the need for a more comprehensive and systematic approach to RM assessment.

**Effective Training towards Reasoning for Reward Modeling.** Our specialized, reasoning-oriented training pipeline delivers substantial performance gains. For instance, RM-R1-QWEN-INSTRUCT-14B consistently surpasses Self-taught-evaluator-llama-3.1-70B, a reasoning model five times its size. The RM-R1 model series also demonstrates impressive results on RM-Bench, exceeding the top-performing baseline by up to 8.7%. On this most reasoning-intensive benchmark, RM-R1-DEEPSEEK-DISTILLED-QWEN-32B establishes a new state-of-the-art. It achieves 91.8% accuracy in math and 74.1% in code, outperforming the previous best models (73% in math and 63% in code) by significant margins. Furthermore, it also records the strongest reasoning performance among our released models on RewardBench. Despite its performance, our Instruct-based models are remarkably data-efficient, reaching competitive performance using only 8.7K examples for distillation—compared to the 800K examples used in training DeepSeek-Distilled (Guo et al., 2025). Overall, our study underscores the significant *potential of directly adapting large reasoning models into highly effective reward models*.

## 4 ANALYSIS

In this section, we present a series of empirical analyses to understand the key ingredients for training effective reasoning reward models. Our analysis spans scaling effects, design decisions, reasoning ablations, and a case study. We also present additional analysis on training dynamics in Section 4.4.

### 4.1 TRAINING RECIPES

We first investigate the key ingredients underlying the successful training of RM-R1. Through a series of ablation studies, we examine our design choices to identify effective strategies for training high-quality reasoning reward models. We compare the following settings: **Cold Start RL**, **Cold Start RL + Rubrics**, **Cold Start RL + Rubrics + Query Categorization (QC)**, and **Distilled + RL + Rubrics + QC** (*i.e.*, **RM-R1**). The details of these settings are in Section I.1.

Table 2: Ablation study of the design choices for Reasoning Training on RewardBench.

| Method | Chat | Chat Hard | Safety | Reasoning | Average |
|---|---|---|---|---|---|
| Instruct (Original) | **95.8** | 74.3 | 86.8 | 86.3 | 85.8 |
| Instruct + **Cold Start RL** | 92.5 | 81.5 | 89.7 | 94.4 | 89.5 |
| Instruct + **Cold Start RL + Rubrics** | 93.0 | 82.5 | 90.8 | 94.2 | 90.1 |
| Instruct + **Cold Start RL + Rubrics + QC** | 92.3 | 82.6 | 91.6 | **96.3** | 90.8 |
| **RM-R1** | 95.3 | **83.1** | **91.9** | 95.2 | **91.4** |

In Table 2, we present the results of the ablation studies described above, using the Qwen-2.5-Instruct-32B model as the Instruct (Original) model. Several key conclusions emerge:

- **RL training alone is insufficient.** While Cold Start RL slightly improves performance on hard chat and reasoning tasks, it fails to close the gap to fully optimized models.

- **CoR prompting optimizes RM rollout and boosts reasoning performance.** Instructing RM-R1 to self-generate chat rubrics or problem solutions before judgment helps overall performance, especially for chat and safety tasks. Incorporating explicit query categorization into the prompt notably improves reasoning performance, suggesting that clearer task guidance benefits learning.

- **Distillation further enhances performance across all axes.** Seeding the model with high-quality reasoning traces before RL yields the strongest results, with improvements observed on both hard tasks and safety-sensitive tasks.

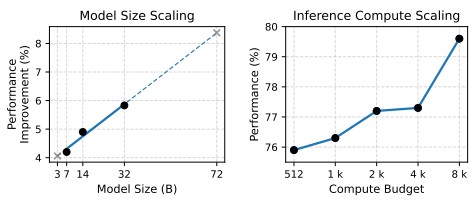

(a) Model Size     (b) Inference Compute

Figure 4: **Scaling effect of RM-R1.** (a) Larger models benefit more from reasoning training. (b) Longer reasoning chains improve RM performance.

Table 3: **Comparison of reasoning-based training versus SFT across benchmarks.** * indicates reasoning-based methods. Reasoning training consistently yields better performance.

| Method | RewardBench | RM-Bench | RMB | Avg. |
|---|---|---|---|---|
| **Train on Full Data** | | | | |
| Instruct + **SFT** | 90.9 | 75.4 | 65.9 | 77.4 |
| Instruct + **Distilled + SFT** | 91.2 | 76.7 | 65.4 | 77.8 |
| RM-R1 * | 91.4 | 79.1 | 73.0 | 81.2 |
| **Train on 9k (Distillation) Data** | | | | |
| Instruct + **SFT** | 88.8 | 74.8 | 66.9 | 76.6 |
| Instruct + **Distilled** * | 89.0 | 76.3 | 72.0 | 79.2 |

## 4.2 SCALING EFFECTS

We then investigate how model performance varies with scale, considering both **model size** and **inference-time compute**. In some cases – such as ScalarRMs from InternLM2 (Cai et al., 2024) and Skywork (Liu et al., 2024) – the smaller models (7B/8B) outperforms the larger ones (20B/27B), showing no advantage of scaling. In this subsection, we show that this trend does not hold for RM-R1, where scaling brings clear and substantial improvements.

### 4.2.1 MODEL SIZES

We first analyze the impact of model scale. Our study is based on the `Qwen-2.5-Instruct` model family at three sizes: 7B, 14B, and 32B. We evaluate performance improvements resulting from our training procedure described in Section 2, with results averaged across three key benchmarks: RewardBench, RM-Bench, and RMB.

For each model size, we compare the original and post-training performance. Figure 4a plots the relative improvement (%) with respect to model size. Observing an approximately linear trend, we fit a linear regression model and extrapolate to hypothetical scales of 3B and 72B, shown using faint markers and dashed extensions. The results strongly support a scaling law for reasoning reward models: larger models not only result in an absolute better final performance but also consistently yield **greater performance gains**. This aligns with the intuition that our training effectively leverages the superior reasoning capabilities of larger models.

### 4.2.2 INFERENCE-TIME COMPUTATION

Next, we examine how model performance varies with different compute budgets measured in number of tokens allowed during inference. Since this is particularly relevant to reasoning-focused models, we fix our base model to `DeepSeek-R1-Distill-Qwen-14B`. We evaluate average performance across the three key benchmarks using a wide range of inference-time compute budgets: 512, 1024, 2048, 4096, and 8192 tokens.

To ensure a fair comparison, we match the training rollout budget to the inference budget in each setting (*i.e.*, we allow a maximum of $k$ tokens during training for a compute budget of $k$ at inference). All models are trained using GRPO with identical datasets and hyperparameter configurations. Figure 4b shows the relationship between compute budget and performance. We observe a clear improvement trend as the inference budget increases. This highlights the benefits of long reasoning chains in reward modeling.

## 4.3 EFFECTIVENESS OF REASONING TRAINING

We now analyze the impact of reasoning-based training. Here, we demonstrate that reasoning-based training can outperform answer-only approaches. We consider the following settings:

**Instruct + SFT.** This approach fine-tunes the instruct model directly toward producing the correct final answer using the full dataset, without providing any intermediate reasoning chains.

**Instruct + Distilled + SFT.** This approach applies SFT (with respect to the answer directly) after the distillation stage, serving as a direct comparison point with RM-R1 trained with RL.

**Instruct + RM-R1 (Distilled + RL).** This is the full approach proposed in this paper, following the procedure detailed in Section 2.

**Instruct + Distilled.** This setting uses the model checkpoint immediately after the distillation stage, before any RL fine-tuning.

In summary, methods with "+ RM-R1" or "+ Distilled" represent reasoning-based approaches, while the remaining methods are purely non-reasoning-based approaches. In Table 3, we report the results measured across the three benchmarks. The findings clearly demonstrate that reasoning training significantly benefits reward model performance. Under fair comparisons (*i.e.*, training on exactly the same amount of data), reasoning-based models consistently outperform their SFT-only counterparts. In particular, even high-quality distillation alone, applied to a small subset of the data, provides notable gains, highlighting the value of incorporating structured intermediate reasoning.

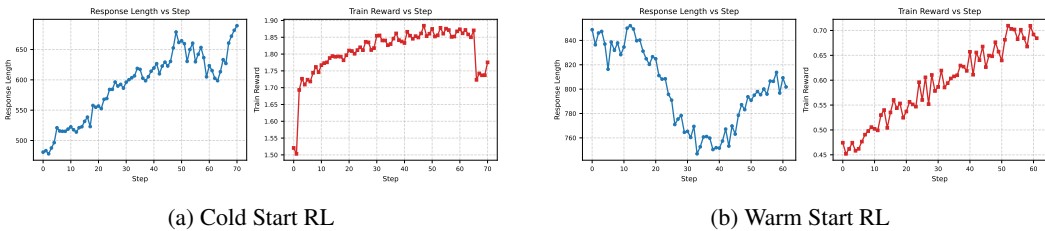

(a) Cold Start RL                    (b) Warm Start RL

Figure 5: RL training dynamics under different settings: (a) Cold Start RL (Eq. 11) and (b) Warm Start RL (Eq. 8). In Cold Start RL, the response length steadily increases as the model learns to reason, but training becomes unstable near the end. In Warm Start RL, the model exhibits more stable training, with effective refinement of reasoning traces throughout the process.

### 4.4 TRAINING DYNAMICS

We analyze the training dynamics of RM-R1 using the `Qwen-2.5-14B-Instruct` model by tracking both response length and reward progression throughout RL training. We consider two settings: (a) Cold Start RL, and (b) Warm Start RL following reasoning-chain distillation. We present the finding in Figure 5.

In the Cold Start RL setting, we observe that the model gradually learns to reason, as reflected by a steady increase in response length over the course of training. However, training becomes unstable near the end, with a sharp drop in the reward curve, suggesting potential issues such as overfitting.

In contrast, under Warm Start RL, the model begins with stronger initial reasoning abilities, exhibiting longer responses from the outset. Interestingly, the model first learns to produce more concise reasoning traces before gradually increasing response length again as training progresses. The reward curve rises smoothly and consistently throughout training, demonstrating more stable and efficient learning compared to the Cold Start setting.

### 4.5 CASE STUDY

To gain deeper insights into RM-R1, we further conduct a case study using `Qwen-14B-Instruct`, comparing its behavior with a cold-start RL model. The results, summarized in Table 4, reveal the following key observations:

**Transparent Judging Process.** The reasoning traces generated by RM-R1 are highly interpretable and coherent, reflecting the model's perception of human preferences. It explicitly articulates why certain responses are better, providing transparency into its evaluation process.

**High-Quality, Question-Dependent Rubrics.** RM-R1 accurately understands the question and the context of comparison, correctly prioritizing "accuracy" as the most critical rubric for medical-related questions. In contrast, the cold-start RL model overlooks the most important factors and

Table 4: **Case Study of RM-R1**(succinct version). Complete Chatbots and model responses are shown in Table 9, 10, and 11. Cold-Start RL evaluation tends to focus on superficial features (*e.g.*, number of suggestions, relevance), whereas RM-R1 produces high-quality rubrics (*e.g.*, accuracy for factual questions), faithfully follows them, and evaluates responses based on their actual content.

---

**Question:** symptoms of sickle-cell disease

---

**Chatbot A:**
1. Painful red or yellow skin lesions; 2. Frequent infections; 3. Fatigue and weakness; 4. Anemia; 5. Slower growth and development; 6. Jaundice; 7. Swelling of hands and feet; 8. Vision loss; 9. Tiredness and weakness after physical activity; 10. Breathlessness; 11. Fever, chills, and night sweats.

**Chatbot B**
Sickle cell disease is a genetic disorder that affects the shape of red blood cells. It is most common in African Americans, but can also affect people from other ethnic groups. The symptoms of sickle cell disease can vary widely from person to person, but some common symptoms include:

1. Fatigue: ...; 2. Pain: ...; 3. Fever: ...; 4. Anemia: ...; 5. Delayed growth and development: ...; 6. Infections: ...; 7. Pigmentation: ...

It's important to note that not everyone with sickle cell disease will experience all of these symptoms, ... Therefore, it's important to consult with a healthcare provider if you suspect you or someone you know may have sickle cell disease.

---

**Cold-start RL evaluation ✗**
`<type>`Chat`</type>`
`<rubric>` **Relevance 40 %** – Comprehensiveness 30 % – Clarity/organisation 30 % `</rubric>`
`<eval>` **Chatbot A provided a list of symptoms that are directly relevant to sickle cell disease.** The response is comprehensive and includes a wide range of symptoms commonly associated with the condition. The symptoms are listed in a clear and organized manner, making it easy for the client to understand the information. Chatbot B's response is less organized than Chatbot A's response, and the explanation of sickle cell disease may be unnecessary for the client's question.`</eval>`
`<answer>`[[A]]`</answer>`

---

**RM-R1 evaluation ✓**
`<type>`Chat`</type>`
`<rubric>` 1 **Accuracy 40 %** · 2 Comprehensiveness 30 % · 3 Clarity 20 % · 4 Helpfulness 10 % `<justify>` Accuracy is weighted highest because providing correct information about medical conditions is crucial. Comprehensiveness is important because ...`</justify>`
`</rubric>`
`<eval>` *Accuracy of information:* Chatbot A provides a list of symptoms without explanation, but **some listed items (e.g. "painful red or yellow skin lesions," "vision loss") are not typical of sickle-cell disease**. Chatbot B offers more accurate information, explaining each symptom's nature and severity. ... *Comprehensiveness:...Clarity and organization:...Helpfulness:...* `</eval>`
`<answer>`[[B]]`</answer>`

---

instead emphasizes superficial or broadly defined features (*e.g.*, relevance). The ability to generate high-quality, question-specific rubrics stems from the knowledge acquired during the distillation.

**Faithful Adherence to Rubrics and Content-Based Judgment.** RM-R1 grounds its evaluation in the actual content of the model responses. It correctly identifies inaccuracies in Chatbot A's response based on factual content rather than surface presentation. Furthermore, it systematically evaluates all aspects of the rubric, leading to a structured, interpretable, and verifiable judging process.

## 5 CONCLUSION AND FUTURE WORK

In this paper, we revisited reward modeling through the lens of reasoning. We introduced RM-R1, a family of REASRMS that effectively generate explicit chains of rubrics and rationales, and scale with both model size and inference compute. Across three public benchmarks, RM-R1 matched or surpassed commercial and open-source RMs while producing more interpretable judgments. Ablation investigations reveal that (1) task-type categorization, (2) bootstrapping from high-quality reasoning traces, and (3) RL fine-tuning are all indispensable. Qualitative analyses further showed that RM-R1 learns to prioritize high-impact rubrics, faithfully follow its own criteria and justify coherently. Future work includes active preference collection, where REASRMS use active learning to query human preference only when the current rubric set is insufficient for a new preference sample. Finally, it would be natural to extend our study to multimodal/agentic reward modeling scenarios.

## ETHICS STATEMENT

RM-R1 focuses on fundamental research in reinforcement learning and reward modeling. Our methods are developed and evaluated entirely on publicly available benchmarks without involving human subjects, sensitive personal data, or private information. The proposed training pipeline is designed as a general optimization technique and does not raise concerns regarding fairness, bias, discrimination, privacy, or security. We believe that our study poses no foreseeable ethical risks and fully complies with research integrity standards.

## REPRODUCIBILITY STATEMENT

We have taken concrete steps to facilitate independent reproduction of our results. The experimental setup, datasets, baselines, and evaluation protocols are detailed in Sections 3, 4 and F.1, with training/evaluation hyperparameters, rollout settings, and compute requirements consolidated in Section G. We provide a supplementary repository (see `https://github.com/RM-R1-UIUC/RM-R1`) containing: (i) Distillation data curation and training from OpenAI and Claude models, (ii) RL training with publicly available preference data, (iii) prompt templates for all settings (Distillation data curation, distillation training and RL training for instruct/distilled models), (iv) the chain-of-rubrics evolution mechanism and implementation details for rubric sampling and aggregation, and (v) the complete evaluation scripts for RewardBench, RMB and RM-Bench. Third-party models and services (e.g., `Llama`, `Qwen`, `DeepSeek-Distilled-Qwen`, `GPT-4o`, `Gemini`) are versioned in the configs. Deviations from defaults and ablations are included with their configs, while more discussions about computational overhead are in Section J. We welcome the reviewers to reproduce our results.

## ACKNOWLEDGEMENT

The authors would like to thank Prof. ChengXiang Zhai for his insightful discussions. This research is based upon work supported by DARPA ITM Program No. FA8650-23-C-7316, the AI Research Institutes program by National Science Foundation and the Institute of Education Sciences, U.S. Department of Education through Award # 2229873 - AI Institute for Transforming Education for Children with Speech and Language Processing Challenges, IBM-Illinois Discovery Accelerator Institute (IIDAI) Center, and NSF Award # 2416070. The views and conclusions contained herein are those of the authors and should not be interpreted as necessarily representing the official policies, either expressed or implied, of the U.S. Government. The U.S. Government is authorized to reproduce and distribute reprints for governmental purposes notwithstanding any copyright annotation therein.

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

CONTENTS

## A    THE USE OF LARGE LANGUAGE MODELS (LLMS)

LLMs did not play a role in shaping the research ideas or writing of this paper to an extent that would merit authorship or contributor status. Within this work, LLMs are regarded strictly as the primary subject of study and serve as the central experimental object in our evaluations.

## B    RELATED WORK

**Reward Models (RMs).**   Early RMs were typically outcome-focused: trained to predict human preference rankings for complete outputs (Zhong et al., 2025). Recent advances have looked at providing process supervision, which rewards or evaluates the steps of a model's reasoning rather than only the final answer. A series of works propose to train process reward models that judge the correctness of intermediate reasoning steps (Lightman et al., 2023; Cui et al., 2025; Setlur et al., 2025). A limitation of many PRMs is their heavy reliance on curated step-level human labels or specific schemas, and they often remain domain-specific. Zhang et al. (2025) propose *Generative Verifiers*, framing reward modeling as a next-token prediction task. This allows the reward model to leverage chain-of-thought and even use majority voting over multiple sampled rationales to make more reliable judgments. DeepSeek-GRM (Liu et al., 2025b) and JudgeLRM (Chen et al., 2025) have studied using reasoning models as generative reward models, which are the most relevant research to ours. However, their main focus is on the effect of scaling inference-time computation on reward modeling. On the contrary, our work is the first to provide a systematic empirical comparison of different reward model training paradigms, shedding light on when and why a distilled and RL-trained reward model like RM-R1 has advantages over the conventional approaches.

**Reinforcement Learning from Human Feedback (RLHF).**   Early works (Christiano et al., 2017) first demonstrated that reinforcement learning could optimize policies using a reward model trained from human pairwise preferences. Subsequent studies applied RLHF to large-scale language models using policy optimization algorithms such as PPO (Schulman et al., 2017). For example, Ziegler et al. (2019) fine-tuned GPT-2 via PPO on human preference rewards, and Stiennon et al. (2020) showed that RLHF could significantly improve the quality of summarization by optimizing against a learned preference model. More recently, Ouyang et al. (2022) used a similar PPO-based pipeline to train InstructGPT, establishing the modern RLHF paradigm for instruction-following models. Recently, Verifiable supervision techniques have also emerged: DeepSeek-R1 (Guo et al., 2025) uses a form of self-verification during RLHF to reward correct reasoning steps, rather than only final-answer quality. This method incentivizes policies to produce outputs that can be verified for correctness, bridging the gap between pure preference-based feedback and ground-truth signals. However, even with such innovations, most RLHF implementations still treat reward modeling and reasoning as separate stages.

## C    USER PROMPT FOR DEEPSEEK-DISTILLED REASONING MODELS

Large reasoning models such as `DeepSeek-R1-distilled` models (Guo et al., 2025) do not have a system prompt, so we show the user prompt for rollouts in Figure 6.

---

**Chain-of-Rubrics (CoR) Rollout for Reasoning Models**

Please act as an impartial judge and evaluate the quality of the responses provided by two AI Chatbots to the Client question displayed below.

... [Pairwise Input Content] ...

Output your final verdict at last by strictly following this format: '<answer>[[A]]</answer>' if Chatbot A is better, or '<answer>[[B]]</answer>' if Chatbot B is better.

---

Figure 6: **The user prompt used for RM-R1 rollout** (for reasoning models).

## D  DETAILS OF REASONING CHAIN GENERATION

We now expand on the **details of generating high-quality reasoning chains**. We first use the same prompt to query `Claude-3.7-Sonnet`, generating initial reasoning traces. However, approximately 25% of these traces are incorrect, primarily on harder chat tasks. To correct these cases, we pass the original prompt, the incorrect trace, and the correct final answer to `OpenAI-O3`, which then generates a corrected reasoning trace aligned with the right answer.

This two-stage process yields a high-quality distillation set. We deliberately choose the order—first `Claude`, then `O3`—based on qualitative observations: `Claude` excels at solving easier tasks and maintaining attention to safety considerations, whereas `O3` performs better on harder tasks but tends to overemphasize helpfulness at the expense of safety. We select approximately 12% of the training data (slightly fewer than 9K examples) for distillation. This is then followed by RL training.

## E  GROUP RELATIVE POLICY OPTIMIZATION (GRPO)

Group Relative Policy Optimization (GRPO) (Shao et al., 2024) is a variant of Proximal Policy Optimization (PPO) (Schulman et al., 2017), which obviates the need for additional value function approximation, and uses the average reward of multiple sampled outputs produced in response to the same prompt as the baseline. More specifically, for each prompt $x$, GRPO samples a group of outputs $\{y_1, y_2, \cdots, y_G\}$ from the old policy $\pi_{\theta_{old}}$ and then optimizes the policy model by maximizing the following objective:

$$
\mathcal{J}_{\text{GRPO}}(\theta) = \mathbb{E}_{x \sim \mathcal{D}, \{j_i\}_{i=1}^{G} \sim r_{\theta_{\text{old}}}(j|x)} \left[ \frac{1}{G} \sum_{i=1}^{G} \frac{1}{|j_i|} \sum_{t=1}^{|j_i|} \left\{ \min \left( \frac{r_\theta(j_{i,t} \mid x, j_{i,<t})}{r_{\theta_{\text{old}}}(j_{i,t} \mid x, j_{i,<t})} \hat{A}_{i,t}, \right. \right. \right.
$$
$$
\left. \left. \left. \text{clip} \left( \frac{r_\theta(j_{i,t} \mid x, j_{i,<t})}{r_{\theta_{\text{old}}}(j_{i,t} \mid x, j_{i,<t})}, 1 - \epsilon, 1 + \epsilon \right) \hat{A}_{i,t} \right) - \beta \, \mathbb{D}_{\text{KL}} \left[ r_\theta(\cdot \mid x) \, \| \, \pi_{\text{ref}}(\cdot \mid x) \right] \right\} \right], \tag{9}
$$

where $\beta$ is a hyperparameter balancing the task specific loss and the KL-divergence. Specifically, $\hat{A}_i$ is computed using the rewards of a group of responses within each group $\{r_1, r_2, \ldots, r_G\}$, and is given by the following equation:

$$
\hat{A}_i = \frac{r_i - \text{mean}(\{r_1, r_2, \cdots, r_G\})}{\text{std}(\{r_1, r_2, \cdots, r_G\})}. \tag{10}
$$

## F  EXPERIMENT SETUPS

### F.1  BENCHMARKS

In this paper, we consider the following three benchmarks:

**RewardBench** (Lambert et al., 2025): RewardBench is one of the first endeavors towards benchmarking reward models through prompt-chosen-rejected trios, covering four categories: chat, chat-hard, reasoning, and safety, with 358, 456, 740, and 1431 samples, respectively.

**RM-Bench** (Liu et al., 2025a): Building on RewardBench, RM-Bench evaluates reward models for their sensitivity to subtle content differences and robustness against style biases. It includes four categories: Chat, Safety, Math, and Code, with 129, 441, 529, and 228 samples, respectively. Each sample contains three prompts of varying difficulty. RM-Bench is the most reasoning-intensive benchmark among those we consider.

**RMB** (Zhou et al., 2025): Compared with RewardBench and RM-Bench, RMB offers a more comprehensive evaluation of helpfulness and harmlessness. It includes over 49 real-world scenarios and supports both pairwise and Best-of-N (BoN) evaluation formats. RMB comprises 25,845 instances in total—37 scenarios under the helpfulness alignment objective and 12 under harmlessness.

### F.2 PREFERENCE DATASETS

We consider the following datasets for training:

**Skywork Reward Preference 80K** (Liu et al., 2024) is a high-quality collection of pairwise preference data drawn from a variety of domains, including chat, safety, mathematics, and code. It employs an advanced data filtering technique to ensure preference reliability across tasks. However, we identify a notable issue with this dataset: all samples from the `magpie_ultra` source exhibit a strong spurious correlation, where rejected responses consistently contain the token "`<im_start>`," while accepted responses do not. Additionally, responses from this source show a systematic bias—accepted responses are typically single-turn, while rejected responses are multi-turn. This problematic subset constitutes approximately 30% of the Skywork dataset and primarily covers mathematics and code domains. To avoid introducing spurious correlations into training, we exclude all `magpie_ultra` data and retain only the cleaned subset for our experiments.

**Code-Preference-Pairs** is a high-quality coding preference dataset. It is constructed by prompting a model with original code, introducing deliberate bugs, and manipulating examples (*e.g.*, swapping broken and corrected versions, removing error comments) to generate fine-grained preference pairs. We subsample 8K examples from this dataset for use in our experiments.

**Math-DPO-10K** (Lai et al., 2024) is a high-quality stepwise preference dataset focused on mathematical reasoning. We use the full dataset in our experiments.

A global statistics of our training dataset is summarized in Table 5.

Table 5: **Global Statistics of our Training Dataset.** * indicates the source is from Skywork-Reward-Preference-80K-v0.2.

| Source | Size | Domain |
|---|---|---|
| magpile_pro_llama3.1* | 29682 | Reasoning |
| offset_bias* | 8504 | Chat (length bias) |
| helpsteer2* | 7221 | Chat |
| wildguard* | 6709 | Safety |
| magpile_pro* | 2030 | Chat |
| Code-Preference-Pairs | 8000 | Reasoning |
| Math-DPO-10K | 10000 | Reasoning |

### F.3 BASELINES

We compare RM-R1 with RMs from three categories:

**ScalarRMs.** ScalarRMs produce a score for model response directly, predicting preference through single numeric scores without explicit reasoning traces. This category includes models such as Eurus-RM (Yuan et al., 2025), Internlm2 (Cai et al., 2024) SteerLM-RM (Wang et al., 2024c), Nemotron-RM (Adler et al., 2024), Tulu-v2.5 (Ivison et al., 2024), Starling-RM (Zhu et al., 2024), ArmoRM (Wang et al., 2024a), Skywork-RM (Liu et al., 2024), etc. While these models often achieve strong results on well-defined benchmarks, they generally lack interpretability and struggle to capture fine-grained reasoning.

**GenRMs.** Generative reward models (GenRMs) offer more expressive feedback by producing free-form textual judgments, typically without further training. This includes the widely used LLM-as-a-Judge setup (Zheng et al., 2023), where pretrained language models are prompted to explain and evaluate responses. We also categorize under GenRMs models that directly generate output answers without intermediate reasoning steps. Representative examples include LLaMA (Dubey et al., 2024), Qwen (Yang et al., 2024), Claude (Anthropic, 2024), GPT-4o (Achiam et al., 2023; Hurst et al., 2024), Gemini 1.5 Pro (Reid et al., 2024), and Skywork-Critic (Shiwen et al., 2024). By leveraging LLMs' generative capabilities, these models enhance interpretability through natural language rationales and explanations.

**REASRMS.** Reasoning-enhanced reward models (REASRMs) explicitly incorporate reasoning processes before their final judgments, often trained through critiques or chain-of-thought strategies.

Notable examples are JudgeLRM (Chen et al., 2025), Critique-RM (Yu et al., 2025), DeepSeek-GRM (Liu et al., 2025b), Self-taught Evaluators (Wang et al., 2024b) and our proposed RM-R1 models. These models excel in tasks demanding rigorous reasoning, safety evaluations, and nuanced preference judgments due to their grounding in structured critical thinking.

## G    IMPLEMENTATION DETAILS

Our training framework is based on VERL (Sheng et al., 2025) and OpenRLHF (Hu et al., 2024). For `Instruct` models, we use 8.7k data for distillation and 64k for RLVR. For `Deepseek-Distilled` models, we use the full data for RLVR.

**Distillation Stage.** We use the `SFTTrainer` from OpenRLHF with a fixed batch size of 128 and a micro-batch size of 1, training for a single epoch. To optimize GPU memory usage, we enable gradient checkpointing, FlashAttention, and Adam offloading. The learning rates are set based on the model size: $5e-6$, $3e-6$, and $2e-6$ for models of size 7B, 14B, and 32B, respectively.

**RLVR Stage.** We use the VERL framework for all GRPO training. The training batch size is fixed at 1024, with a mini-batch size of 128. We adopt Fully Sharded Data Parallel (FSDP) to improve memory efficiency. For rollout generation, we use vLLM with tensor parallelism size 4 and GPU memory utilization capped at 0.4. Sampling follows default parameters (temperature = 1.0, top-p = 1.0). KL regularization is applied with a coefficient of $1e-3$ and a clip ratio of 0.2. Each prompt is sampled with 7 candidate responses.

The maximum input sequence length is 4,096 tokens, and the maximum response length is 8,192 tokens. Learning rates are set separately for the two model variants:

- Instruct models: $1e-6$, $7e-7$, and $5e-7$ for 7B, 14B, and 32B models, respectively.
- Reasoning models: $1e-6$, $1e-6$, and $8e-7$ for 7B, 14B, and 32B models, respectively.

We train the 7B, 14B, and 32B models on 1, 2, and 4 nodes, respectively, each equipped with 8 H100 GPUs.

## H    FULL EXPERIMENT RESULT

In this section, we provide the full experiment results and a more comprehensive coverage of existing baselines. The results of RewardBench, RM-Bench, and RMB are provided in Table 6, Table 7, Table 8, respectively.

## I    SUPPLEMENTARY INFORMATION FOR SECTION 4

### I.1    ABLATION SETTINGS

**Cold Start RL.** This approach generally involves pure RL, with rule-based rewards centered on answer correctness and format compliance. Such strategies have achieved notable success in advanced mathematical problem solving (Shao et al., 2024).

In this setting, we replicate this conventional training setup. Specifically, we use a combination of a format reward and an answer reward:

$$\mathcal{R}_{\text{format}} = \begin{cases} 1 & \text{if format matches,} \\ 0 & \text{otherwise,} \end{cases} \quad \text{and} \quad \mathcal{R}_{\text{answer}} = \begin{cases} 1 & \text{if answer matches,} \\ 0 & \text{otherwise.} \end{cases} \tag{11}$$

The total reward is the sum $\mathcal{R} = \mathcal{R}_{\text{answer}} + \mathcal{R}_{\text{format}}$. We use the prompt template shown in Figure 8, a version without any guidance on structured reasoning.

**Cold Start RL + Rubrics.** To examine the influence of structured reasoning in final model performance, compared with the last setting, we use the prompt template shown in Figure 7. Compared with

Table 6: Results of our proposed method and baselines on the RewardBench. **Bold** numbers indicate the best performance, Underlined numbers indicate the second best. ✥ indicates potential data contamination.

| Models | Chat | Chat_Hard | Safety | Reasoning | Overall |
|---|---|---|---|---|---|
| ***ScalarRMs*** | | | | | |
| Eurus-RM-7b | 98.0 | 65.6 | 81.4 | 86.3 | 82.8 |
| Internlm2-7b-reward | 99.2 | 69.5 | 87.2 | 94.5 | 87.6 |
| SteerLM-RM 70B | 91.3 | 80.3 | 92.8 | 90.6 | 88.8 |
| Cohere-0514 | 96.4 | 71.3 | 92.3 | 97.7 | 89.4 |
| Internlm2-20b-reward | 98.9 | 76.5 | 89.5 | 95.8 | 90.2 |
| ArmoRM-Llama3-8B-v0.1 | 96.9 | 76.8 | 90.5 | 97.3 | 90.4 |
| Nemotron-4-340B-Reward | 95.8 | **87.1** | 91.5 | 93.6 | 92.0 |
| Skywork-Reward-Llama-3.1-8B✥ | 95.8 | 87.3 | 90.8 | 96.2 | 92.5 |
| Skywork-Reward-Gemma-2-27B✥ | 95.8 | 91.4 | 91.9 | 96.1 | 93.8 |
| INF-ORM-Llama3.1-70B | 96.6 | 91.0 | **93.6** | **99.1** | **95.1** |
| ***GenRMs*** | | | | | |
| Llama3.1-8B-Instruct | 85.5 | 48.5 | 75.6 | 72.1 | 70.4 |
| Prometheus-8*7B-v2 | 93.0 | 47.1 | 80.5 | 77.4 | 74.5 |
| Llama3.1-70B-Instruct | **97.2** | 70.2 | 82.8 | 86.0 | 84.0 |
| Llama3.1-405B-Instruct | **97.2** | 74.6 | 77.6 | 87.1 | 84.1 |
| Claude-3-5-sonnet-20240620 | 96.4 | 74.0 | 81.6 | 84.7 | 84.2 |
| GPT-4o-0806 | 96.1 | 76.1 | 86.6 | 88.1 | 86.7 |
| Gemini-1.5-pro | 92.3 | 80.6 | 87.9 | 92.0 | 88.2 |
| SFR-LLaMa-3.1-70B-Judge-r | 96.9 | 84.8 | 91.6 | 97.6 | 92.7 |
| Skywork-Critic-Llama-3.1-70B✥ | 96.6 | 87.9 | 93.1 | 95.5 | 93.3 |
| **REASRMs** | | | | | |
| JudgeLRM | 92.9 | 56.4 | 78.2 | 73.6 | 75.2 |
| SynRM | 38.0 | 82.5 | 74.1 | 87.1 | 70.4 |
| RM-R1-DEEPSEEK-DISTILLED-QWEN-7B | 88.9 | 66.2 | 78.4 | 87.0 | 80.1 |
| CLoud | 97.0 | 58.0 | 84.0 | 92.0 | 82.8 |
| DeepSeek-GRM-16B | 90.8 | 74.3 | 84.7 | 81.8 | 82.9 |
| DeepSeek-GRM-27B-RFT | 94.7 | 77.2 | 87.0 | 79.2 | 84.5 |
| RM-R1-QWEN-INSTRUCT-7B | 94.1 | 74.6 | 85.2 | 86.7 | 85.2 |
| DeepSeek-GRM-27B | 94.1 | 78.3 | 88.0 | 83.8 | 86.0 |
| DeepSeek-PairRM-27B | 95.5 | 86.8 | 52.3 | 92.0 | 87.1 |
| RM-R1-QWEN-INSTRUCT-14B | 93.6 | 80.5 | 86.9 | 92.0 | 88.2 |
| RM-R1-DEEPSEEK-DISTILLED-QWEN-14B | 91.3 | 79.4 | 89.3 | 95.5 | 88.9 |
| Self-taught-evaluator-llama3.1-70B | 96.9 | 85.1 | 89.6 | 88.4 | 90.0 |
| RM-R1-DEEPSEEK-DISTILLED-QWEN-32B | 95.3 | 80.3 | 91.1 | 96.8 | 90.9 |
| RM-R1-QWEN-INSTRUCT-32B | 95.3 | 83.1 | 91.9 | 95.2 | 91.4 |

the last setting, the model is prompted to generate rubrics and evaluate accordingly. However, compared with the final system prompt of RM-R1 Figure 3, all input prompts are treated uniformly—that is, chat and reasoning tasks are not distinguished.

**Cold Start RL + Rubrics + Query Categorization (QC).** This setting largely follows the previous one, with a key modification: prompting the LM to first categorize the task into reasoning or chat, and then apply different strategies for handling those tasks. Intuitively, reinforcement learning alone can effectively explore reasoning tasks, a domain where it has already achieved considerable success. Here, we incorporate the system prompt shown in Figure 3, which explicitly distinguishes between chat and reasoning tasks.

For reasoning tasks specifically, we note that answer quality is closely tied to correctness, and that high-level rubrics may be less effective than simply evaluating whether the model can solve the problem and verify its own answer. Thus, this setting emphasizes correctness-based evaluation guided by task classification in the prompt.

**Distilled + RL + Rubrics + QC (RM-R1).** Building on the previous setup, we introduce an additional distillation stage from stronger teacher models as a warm start before RL training. The motivation is that with RL alone, weaker models (especially at smaller scales) often **fail to explore high-quality rubrics** and convincing reasoning chains for chat tasks throughout the RL training process. Distilling strong reasoning traces on a small subset of data can effectively mitigate this limitation.

Table 7: The full results of tested reward models on RM-Bench. Chat, Math, Code, Safety show the model's Average Accuracy on each domain. Easy, Normal, Hard show the model's Accuracy on each difficulty level across all domains. **Bold** numbers indicate the best performance, Underlined numbers indicate the second best.

| Models | Chat | Math | Code | Safety | Easy | Normal | Hard | Avg |
|---|---|---|---|---|---|---|---|---|
| ***ScalarRMs*** | | | | | | | | |
| steerlm-70b | 56.4 | 53.0 | 49.3 | 51.2 | 48.3 | 54.9 | 54.3 | 52.5 |
| tulu-v2.5-70b-preference-mix-rm | 58.2 | 51.4 | 55.5 | 87.1 | 72.8 | 65.6 | 50.7 | 63.0 |
| Mistral-7B-instruct-Unified-Feedback | 56.5 | 58.0 | 51.7 | 86.8 | 87.1 | 67.3 | 35.3 | 63.2 |
| RM-Mistral-7B | 57.4 | 57.0 | 52.7 | 87.2 | 88.6 | 67.1 | 34.9 | 63.5 |
| Eurus-RM-7b | 59.9 | 60.2 | 56.9 | 86.5 | 87.2 | 70.2 | 40.2 | 65.9 |
| internlm2-7b-reward | 61.7 | 71.4 | 49.7 | 85.5 | 85.4 | 70.7 | 45.1 | 67.1 |
| Skywork-Reward-Gemma-2-27B | 69.5 | 54.7 | 53.2 | 91.9 | 78.0 | 69.2 | 54.9 | 67.3 |
| ArmoRM-Llama3-8B-v0.1 | 67.8 | 57.5 | 53.1 | 92.4 | 82.2 | 71.0 | 49.8 | 67.7 |
| GRM-llama3-8B-sftreg | 62.7 | 62.5 | 57.8 | 90.0 | 83.5 | 72.7 | 48.6 | 68.2 |
| internlm2-20b-reward | 63.1 | 66.8 | 56.7 | 86.5 | 82.6 | 71.6 | 50.7 | 68.3 |
| Llama-3-OffsetBias-RM-8B | 71.3 | 61.9 | 53.2 | 89.6 | 84.6 | 72.2 | 50.2 | 69.0 |
| Nemotron-340B-Reward | 71.2 | 59.8 | 59.4 | 87.5 | 81.0 | 71.4 | 56.1 | 69.5 |
| URM-LLaMa-3.1-8B | 71.2 | 61.8 | 54.1 | 93.1 | 84.0 | 73.2 | 53.0 | 70.0 |
| Skywork-Reward-Llama-3.1-8B | 69.5 | 60.6 | 54.5 | **95.7** | 89.0 | 74.7 | 46.6 | 70.1 |
| INF-ORM-Llama3.1-70B | 66.3 | 65.6 | 56.8 | 94.8 | **91.8** | 76.1 | 44.8 | 70.9 |
| ***GenRMs*** | | | | | | | | |
| tulu-v2.5-dpo-13b-chatbot-arena-2023 | 64.9 | 52.3 | 50.5 | 62.3 | 82.8 | 60.2 | 29.5 | 57.5 |
| tulu-v2.5-dpo-13b-nectar-60k | 56.3 | 52.4 | 52.6 | 73.8 | 86.7 | 64.3 | 25.4 | 58.8 |
| stablelm-2-12b-chat | 67.2 | 54.9 | 51.6 | 65.2 | 69.1 | 63.5 | 46.6 | 59.7 |
| tulu-v2.5-dpo-13b-stackexchange-60k | 66.4 | 49.9 | 54.2 | 69.0 | 79.5 | 63.0 | 37.2 | 59.9 |
| Nous-Hermes-2-Mistral-7B-DPO | 58.8 | 55.6 | 51.3 | 73.9 | 69.5 | 61.1 | 49.1 | 59.9 |
| Claude-3-5-sonnet-20240620 | 62.5 | 62.6 | 54.4 | 64.4 | 73.8 | 63.4 | 45.9 | 61.0 |
| tulu-v2.5-dpo-13b-hh-rlhf-60k | 68.4 | 51.1 | 52.3 | 76.5 | 53.6 | 63.0 | 69.6 | 62.1 |
| tulu-2-dpo-13b | 66.4 | 51.4 | 51.8 | 85.4 | 86.9 | 66.7 | 37.7 | 63.8 |
| SOLAR-10.7B-Instruct-v1.0 | **78.6** | 52.3 | 49.6 | 78.9 | 57.5 | 67.6 | 69.4 | 64.8 |
| Llama3.1-70B-Instruct | 64.3 | 67.3 | 47.5 | 83.0 | 74.7 | 67.8 | 54.1 | 65.5 |
| Skywork-Critic-Llama-3.1-70B | 71.4 | 64.6 | 56.8 | 94.8 | 85.6 | 73.7 | 56.5 | 71.9 |
| GPT-4o-0806 | 67.2 | 67.5 | 63.6 | 91.7 | 83.4 | 75.6 | 58.7 | 72.5 |
| Gemini-1.5-pro | 71.6 | 73.9 | 63.7 | 91.3 | 83.1 | 77.6 | 64.7 | 75.2 |
| **REASRMs** | | | | | | | | |
| JudgeLRM | 59.9 | 59.9 | 51.9 | 87.3 | 73.2 | 766.2 | 54.8 | 64.7 |
| RM-R1-QWEN-INSTRUCT-7B | 66.6 | 67.0 | 54.6 | 92.6 | 79.2 | 71.7 | 59.7 | 70.2 |
| Self-taught-evaluator-llama3.1-70B | 73.4 | 65.7 | 56.3 | 90.4 | 80.2 | 74.5 | 59.7 | 71.5 |
| RM-R1-DEEPSEEK-DISTILLED-QWEN-7B | 64.0 | 83.9 | 56.2 | 85.3 | 75.9 | 73.1 | 68.1 | 72.4 |
| RM-R1-QWEN-INSTRUCT-14B | 75.6 | 75.4 | 60.6 | 93.6 | 82.6 | 77.5 | 68.8 | 76.1 |
| RM-R1-QWEN-INSTRUCT-32B | 75.3 | 80.2 | 66.8 | 93.9 | 86.3 | 80.5 | 70.4 | 79.1 |
| RM-R1-DEEPSEEK-DISTILLED-QWEN-14B | 71.8 | 90.5 | 69.5 | 94.1 | 86.2 | 83.6 | 74.4 | 81.5 |
| RM-R1-DEEPSEEK-DISTILLED-QWEN-32B | 74.2 | **91.8** | **74.1** | 95.4 | 89.5 | **85.4** | **76.7** | **83.9** |

## J  COMPUTATIONAL OVERHEAD

The core contribution of RM-R1 lies in re-casting reward modeling as a reasoning task and demonstrating, for the first time, that generative RMs can outperform scalar RMs on public benchmarks with fully transparent training recipes and detailed, from-scratch analysis. While long-chain-of-thought models naturally incur higher inference cost, they offer substantial gains in interpretability, generalization (Merrill & Sabharwal; Li et al.), and broader applicability (Gunjal et al., 2025; Fernández-Sánchez et al., 2025). This is analogous to the introduction of DeepSeek-R1, which showcased the potential of reasoning models before subsequent work improved efficiency. We believe REASRMs will become more important in the future because it can dynamically allocate more compute to more complex problems, which cannot be done in conventional RMs. We similarly view efficiency improvements as orthogonal to our main contribution.

While the long chain-of-thought outputs of REASRMs naturally increase inference latency, this overhead can be mitigated with modern RL engines and careful system design. In conventional RL pipelines, the rollout and reward computation stages are often cascaded, leading to a total wait time that is the sum of both processes. However, a parallel design can be implemented where the next batch of rollouts is initiated while the reward model is processing the current batch. In such a setup,

Table 8: The leaderboard of RMB, ranked by the average score of all subsets. **Bold** numbers indicate the best performance, Underlined numbers indicate the second best.

| Models | Helpfulness | | Harmlessness | | Overall |
|---|---|---|---|---|---|
| | BoN | Pairwise | BoN | Pairwise | |
| *ScalarRMs* | | | | | |
| Tulu-v2.5-13b-preference-mix-rm | 0.355 | 0.562 | 0.351 | 0.545 | 0.453 |
| SteerLM-RM 70B | 0.502 | 0.574 | 0.578 | 0.673 | 0.582 |
| Skywork-Reward-Gemma-2-27B | 0.472 | 0.653 | 0.561 | 0.721 | 0.602 |
| Internlm2-20b-reward | 0.585 | 0.763 | 0.499 | 0.670 | 0.629 |
| ArmoRM-Llama3-8B-v0.1 | 0.636 | 0.787 | 0.497 | 0.663 | 0.646 |
| Internlm2-7b-reward | 0.626 | 0.782 | 0.563 | 0.712 | 0.671 |
| Eurus-RM-7b | 0.679 | 0.818 | 0.543 | 0.693 | 0.683 |
| Skywork-Reward-Llama-3.1-8B | 0.627 | 0.781 | 0.603 | 0.759 | 0.693 |
| INF-ORM-Llama3.1-70B | 0.650 | 0.798 | 0.607 | 0.767 | 0.705 |
| Starling-RM-34B | 0.604 | 0.774 | 0.674 | 0.795 | 0.712 |
| *GenRMs* | | | | | |
| Llama2-70b-chat | 0.289 | 0.613 | 0.249 | 0.602 | 0.438 |
| Llama3.1-8B-Instruct | 0.365 | 0.675 | 0.267 | 0.653 | 0.490 |
| Gemini-1.5-pro | 0.536 | 0.763 | 0.299 | 0.661 | 0.565 |
| Mixtral-8x7B-Instruct-v0.1 | 0.480 | 0.706 | 0.491 | 0.671 | 0.587 |
| skywork-critic-llama3.1-8B | 0.600 | 0.725 | 0.578 | 0.578 | 0.620 |
| skywork-critic-llama3.1-70B | 0.640 | 0.753 | 0.614 | 0.614 | 0.655 |
| Llama3.1-70B-Instruct | 0.648 | 0.811 | 0.558 | 0.739 | 0.689 |
| Mistral-Large-2407 | 0.678 | 0.817 | 0.583 | 0.725 | 0.701 |
| Claude-3-5-sonnet | **0.705** | **0.838** | 0.518 | 0.764 | 0.706 |
| Qwen2-72B-Instruct | 0.645 | 0.810 | 0.649 | 0.789 | 0.723 |
| GPT-4o-2024-05-13 | 0.639 | 0.815 | **0.682** | **0.814** | **0.738** |
| **REASRMs** | | | | | |
| JudgeLRM | 0.363 | 0.699 | 0.363 | 0.674 | 0.531 |
| RM-R1-DEEPSEEK-DISTILLED-QWEN-7B | 0.451 | 0.658 | 0.429 | 0.664 | 0.551 |
| RM-R1-QWEN-INSTRUCT-7B | 0.543 | 0.740 | 0.608 | 0.765 | 0.664 |
| Self-taught-evaluator-llama3.1-70B | 0.616 | 0.786 | 0.546 | 0.733 | 0.670 |
| Deepseek-GRM-27B-RFT | 0.592 | 0.801 | 0.548 | 0.765 | 0.670 |
| RM-R1-DEEPSEEK-DISTILLED-QWEN-14B | 0.593 | 0.765 | 0.613 | 0.769 | 0.685 |
| Deepseek-GRM-27B | 0.623 | 0.805 | 0.570 | 0.761 | 0.690 |
| RM-R1-QWEN-INSTRUCT-14B | 0.594 | 0.776 | 0.620 | 0.778 | 0.692 |
| RM-R1-DEEPSEEK-DISTILLED-QWEN-32B | 0.620 | 0.782 | 0.618 | 0.771 | 0.698 |
| RM-R1-QWEN-INSTRUCT-32B | 0.636 | 0.791 | **0.682** | 0.809 | 0.730 |

the total time is determined by the maximum of the two stages' latencies, not their sum. Since the policy model's rollout time (T1) and the reward model's computation time (T2) are often similar for complex tasks, this parallelism can make the practical overhead of using a REASRM minimal.

**Chain-of-Rubrics (CoR) Roll-out for Instruct Models**
**(no categorization of task types)**

Please act as an impartial judge and evaluate the quality of the responses provided by two AI Chatbots to the Client's question displayed below.

**Instructions**
1. Begin your evaluation by generating the rubric criteria tailored to the Client's question and context.
   Enclose the rubric in <rubric> ... </rubric> tags.
2. Assign weights to each rubric item based on their relative importance.
3. Within <rubric>, include a <justify> ... </justify> section explaining the rationale behind the chosen criteria and weights.
4. Compare both Chatbot responses using the rubric.
5. Include your evaluation in <eval> ... </eval> tags.
   Support your analysis using:
     - <quote_A> ... </quote_A> for direct quotes from Chatbot A
     - <summary_A> ... </summary_A> for paraphrased summaries of Chatbot A
     - <quote_B> ... </quote_B> for direct quotes from Chatbot B
     - <summary_B> ... </summary_B> for paraphrased summaries of Chatbot B
6. Conclude with your final judgment using:
   <answer>[[A]]</answer>    or    <answer>[[B]]</answer>

**Important Notes:**
- Be objective and base your evaluation strictly on the content of the responses.
- Do not let the response order, length, or Chatbot names bias your judgment.

Figure 7: The system prompt of the ablation study on cold start RL without categorization of task types.

**Chain-of-Rubrics (CoR) Roll-out for Instruct Models**
**(no rubrics)**

Please act as an impartial judge and evaluate the quality of the responses provided by two AI Chatbots to the Client's question displayed below.

You should choose the chatbot that follows the client's instructions and answers the client's question better. Do not allow the length of the responses to influence your evaluation. Do not favor certain names of the chatbots. Be as objective as possible. First, compare the chatbot responses and provide your evaluations. Then, conclude with your verdict using exactly this format: <answer>[[A]]</answer> if Chatbot A is better, <answer>[[B]]</answer> if Chatbot B is better.

Figure 8: The system prompt of the ablation study on cold start RL without any rubrics.

Table 9: Complete questions and chatbot responses.

| |
|---|
| **Question:** symptoms of sickle-cell disease |
| **Chatbot A:**
1. Painful red or yellow skin lesions
2. Frequent infections
3. Fatigue and weakness
4. Anemia
5. Slower growth and development
6. Jaundice
7. Swelling of hands and feet
8. Vision loss
9. Tiredness and weakness after physical activity
10. Breathlessness
11. Fever, chills, and night sweats.

**Chatbot B**
Sickle cell disease is a genetic disorder that affects the shape of red blood cells. It is most common in African Americans, but can also affect people from other ethnic groups. The symptoms of sickle cell disease can vary widely from person to person, but some common symptoms include:

1. Fatigue: People with sickle cell disease often experience fatigue and weakness, even when they have not been physically active.
2. Pain: Sickle cell disease can cause pain in the body, particularly in the arms, legs, back, and abdomen. Pain can be mild or severe and can last for days or weeks.
3. Fever: People with sickle cell disease may be prone to infections and may experience frequent fevers.
4. Anemia: Sickle cell disease can cause anemia, a condition in which the body does not have enough red blood cells to carry oxygen throughout the body.
5. Delayed growth and development: Children with sickle cell disease may be smaller than their peers due to delayed growth and development.
6. Infections: People with sickle cell disease are at increased risk for infections, particularly in the bloodstream and urinary tract.
7. Pigmentation: Sickle cell disease can cause dark areas of pigmentation in the body, particularly on the hands and feet.

It's important to note that not everyone with sickle cell disease will experience all of these symptoms, and some people may not experience any symptoms at all. Therefore, it's important to consult with a healthcare provider if you suspect you or someone you know may have sickle cell disease. |

Table 10: Cold-Start RL Judges. The rubrics and evaluation mainly center around superficial shallow features (*e.g.*, number of points listed) without actually taking their accuracy into account.

---

**Cold-start RL evaluation**
<type>Chat</type>
<rubric>
The evaluation criteria for the responses are as follows:
- <item>Relevance to the question</item> (40%): The response should directly address the symptoms of sickle cell disease.
- <item>Comprehensiveness</item> (30%): The response should include a wide range of symptoms commonly associated with sickle cell disease.
- <item>Clarity and organization</item> (30%): The response should be easy to read and understand, with symptoms listed in a clear and organized manner.
<justify>
The question specifically asks for the symptoms of sickle cell disease, so the responses should be directly relevant to this topic. Comprehensiveness is important because sickle cell disease can have a variety of symptoms, and a comprehensive list will provide a more complete picture of the condition. Clarity and organization are important because the symptoms can be complex, and a well-organized response will make it easier for the client to understand the information.
</justify>
</rubric>

<eval>
<quote_A>
1. Painful red or yellow skin lesions
2. Frequent infections
3. Fatigue and weakness
4. Anemia
5. Slower growth and development
6. Jaundice
7. Swelling of hands and feet
8. Vision loss
9. Tiredness and weakness after physical activity
10. Breathlessness
11. Fever, chills, and night sweats.
</quote_A>
<summary_A>
Chatbot A provided a list of symptoms that are directly relevant to sickle cell disease. The response is comprehensive and includes a wide range of symptoms commonly associated with the condition. The symptoms are listed in a clear and organized manner, making it easy for the client to understand the information.
</summary_A>
<quote_B>
Sickle cell disease is a genetic disorder that affects the shape of red blood cells. It is most common in African Americans, but can also affect people from other ethnic groups. The symptoms of sickle cell disease can vary widely from person to person, but some common symptoms include:
1. Fatigue: People with sickle cell disease often experience fatigue and weakness, even when they have not been physically active.
2. Pain: Sickle cell disease can cause pain in the body, particularly in the arms, legs, back, and abdomen. Pain can be mild or severe and can last for days or weeks.
3. Fever: People with sickle cell disease may be prone to infections and may experience frequent fevers.
4. Anemia: Sickle cell disease can cause anemia, a condition in which the body does not have enough red blood cells to carry oxygen throughout the body.
5. Delayed growth and development: Children with sickle cell disease may be smaller than their peers due to delayed growth and development.
6. Infections: People with sickle cell disease are at increased risk for infections, particularly in the bloodstream and urinary tract.
7. Pigmentation: Sickle cell disease can cause dark areas of pigmentation in the body, particularly on the hands and feet.
</quote_B>
<summary_B>
Chatbot B provided a detailed explanation of sickle cell disease and a list of symptoms. The response is relevant to the question and includes a wide range of symptoms commonly associated with the condition. However, the response is less organized than Chatbot A's response, and the explanation of sickle cell disease may be unnecessary for the client's question.
</summary_B>
</eval>

<answer>[[A]]</answer>

---

Table 11: RM-R1 RL Judges. The rubrics and evaluation are well-designed and reasonable, assessing the effectiveness of the model response.

---

**Warm-start RL (RM-R1) evaluation**

<type>Chat</type>

<rubric>

1. Accuracy of information (40 %): How correctly does the response describe the symptoms of sickle-cell disease?

2. Comprehensiveness (30 %): How thoroughly does the response cover the range of symptoms associated with sickle-cell disease?

3. Clarity and organization (20 %): How well-structured and easy to understand is the information presented?

4. Helpfulness for the user (10 %): How useful is the information for someone seeking to understand sickle-cell disease symptoms?

<justify>

These criteria focus on evaluating how well each response addresses the user's question about sickle-cell disease symptoms. Accuracy is weighted highest because providing correct information about medical conditions is crucial. Comprehensiveness is important, as sickle-cell disease has multiple symptoms that vary in severity and presentation. Clarity and organization affect how easily the user can understand the information. Helpfulness considers how practical and actionable the information is for the user.

</justify>

</rubric>

<eval>

**Comparing the two responses:**

*Accuracy of information:*

<quote_A>1. Painful red or yellow skin lesions

2. Frequent infections

3. Fatigue and weakness

4. Anemia

5. Slower growth and development

6. Jaundice

7. Swelling of hands and feet

8. Vision loss

9. Tiredness and weakness after physical activity

10. Breathlessness

11. Fever, chills, and night sweats.</quote_A>

<quote_B>1. Fatigue: People with sickle-cell disease often experience fatigue and weakness, even when they have not been physically active.

2. Pain: Sickle-cell disease can cause pain in the body, particularly in the arms, legs, back, and abdomen. Pain can be mild or severe and can last for days or weeks.

3. Fever: People with sickle-cell disease may be prone to infections and may experience frequent fevers.

4. Anemia: Sickle-cell disease can cause anemia, a condition in which the body does not have enough red blood cells to carry oxygen throughout the body.

5. Delayed growth and development: Children with sickle-cell disease may be smaller than their peers due to delayed growth and development.

6. Infections: People with sickle-cell disease are at increased risk for infections, particularly in the bloodstream and urinary tract.

7. Pigmentation: Sickle-cell disease can cause dark areas of pigmentation in the body, particularly on the hands and feet.</quote_B>

Chatbot A provides a list of symptoms without explanation, but some listed items (e.g. "painful red or yellow skin lesions," "vision loss") are not typical of sickle-cell disease. Chatbot B offers more accurate information, explaining each symptom's nature and severity.

*Comprehensiveness:*

Chatbot A lists 11 symptoms, several of which are inaccurate or uncommon. Chatbot B lists 7 symptoms that are more representative and supplies contextual detail about variability and medical advice.

*Clarity and organization:*

Chatbot A's plain numbered list lacks context, making severity unclear. Chatbot B structures the list with explanations, improving readability and understanding.

*Helpfulness:*

Chatbot A is less helpful due to inaccuracies and missing context. Chatbot B is more useful, providing accurate descriptions, context, and guidance.

</eval>

<answer>[[B]]</answer>

---

## K    ANALYSIS FOR THE IMPORTANCE OF RL

**Lemma 1** (High-reward filtering induces $\varepsilon_{\mathrm{train}} < \delta$). *Let $\mathcal{X}$ be a finite input (or trajectory) space and let $\mu$ be a probability distribution on $\mathcal{X}$ (the underlying environment distribution).*

*Let $\phi_{\mathrm{rob}}, \phi_{\mathrm{triv}} : \mathcal{X} \to \{0, 1\}$ be a robust and a trivial feature, and define the* disagreement event

$$D \; := \; \{x \in \mathcal{X} : \phi_{\mathrm{rob}}(x) \neq \phi_{\mathrm{triv}}(x)\}.$$

*Let $R : \mathcal{X} \to \mathbb{R}$ be a reward (or quality) function such that larger $R(x)$ indicates better performance. Fix a threshold $\tau \in \mathbb{R}$ and define the* high-reward *and* low-reward *events*

$$H \; := \; \{x : R(x) \geq \tau\}, \qquad L \; := \; \{x : R(x) < \tau\}.$$

*Assume:*

  *(i) The SFT training distribution is the environment distribution conditioned on high reward:*

$$p_{\mathrm{train}}(\cdot) \; := \; \mu(\cdot \mid H), \qquad p_{\mathrm{env}}(\cdot) \; := \; \mu(\cdot).$$

  *(ii) The filter is nontrivial:*

$$0 < \mu(H) < 1, \quad equivalently \quad 0 < \mu(L) < 1.$$

  *(iii) Disagreement is more prevalent among low-reward trajectories than among high-reward ones:*

$$\Pr_{\mu}[D \mid L] \; > \; \Pr_{\mu}[D \mid H].$$

*Define*

$$\varepsilon_{\mathrm{train}} \; := \; \Pr_{x \sim p_{\mathrm{train}}}[D] = \Pr_{\mu}[D \mid H], \qquad \delta \; := \; \Pr_{x \sim p_{\mathrm{env}}}[D] = \Pr_{\mu}[D].$$

*Then $\varepsilon_{\mathrm{train}} < \delta$.*

**Remark 1** (Connection to SFT distillation and RL exploration). *Lemma 1 formalizes a realistic pipeline:*

  • *The SFT (or distillation) dataset is collected by sampling trajectories from some base policy or environment, and then* filtering *to keep only those with sufficiently high reward (e.g., judged as "good" by humans or a reward model). This corresponds to $p_{\mathrm{train}} = \mu(\cdot \mid H)$.*

  • *RL, in contrast, optimizes expected reward under the environment distribution $p_{\mathrm{env}} = \mu$, and through on-policy rollouts can visit both high- and low-reward regions $H$ and $L$.*

*If disagreement between trivial and robust features tends to* hurt *reward (i.e., is more common in $L$ than in $H$), then assumption* (iii) *is natural. The lemma then shows that the disagreement probability $\varepsilon_{\mathrm{train}}$ seen in the filtered SFT data is strictly smaller than the true environment disagreement $\delta$.*

*In practice, the gap can be amplified further by sampling effects. Let $N$ be the number of (independent) trajectories in the SFT dataset drawn from $p_{\mathrm{train}}$ and $M$ the number of trajectories seen during RL rollouts drawn from $p_{\mathrm{env}}$. If $\Pr_{p_{\mathrm{train}}}[D] = \varepsilon_{\mathrm{train}}$ is very small, the probability that the SFT dataset contains* no *disagreement example is*

$$\Pr[no \; D \; in \; SFT \; data] = (1 - \varepsilon_{\mathrm{train}})^N,$$

*which can be close to $1$ for moderate $N$. In contrast, the probability that RL ever encounters a disagreement trajectory is*

$$\Pr[at \; least \; one \; D \; in \; RL \; rollouts] = 1 - (1 - \delta)^M,$$

*which quickly approaches $1$ when $M$ is large and $\delta > 0$. Since the number of possible trajectories typically grows exponentially with horizon, a finite distilled SFT dataset can easily miss rare but systematically harmful disagreement patterns, while RL exploration eventually uncovers them. This provides a concrete justification for the core assumption $\varepsilon_{\mathrm{train}} < \delta$ that we use in the next proposition.*

**Proposition 1** (Approximate shortcut agreement on SFT vs. RL). *Let $\mathcal{X}$ be a finite input space and let $p_{\text{train}}$ and $p_{\text{env}}$ be two probability distributions on $\mathcal{X}$.*

*Let*

$$\phi_{\text{rob}}, \phi_{\text{triv}} : \mathcal{X} \to \{0, 1\}$$

*denote a* robust *and a* trivial *feature, respectively, and define the ground-truth label as*

$$y(x) := \phi_{\text{rob}}(x), \qquad x \in \mathcal{X}.$$

*Define the disagreement probabilities*

$$\varepsilon_{\text{train}} := \Pr_{x \sim p_{\text{train}}} \big[ \phi_{\text{triv}}(x) \neq \phi_{\text{rob}}(x) \big], \qquad \delta := \Pr_{x \sim p_{\text{env}}} \big[ \phi_{\text{triv}}(x) \neq \phi_{\text{rob}}(x) \big].$$

*Assume the gap*

$$0 \leq \varepsilon_{\text{train}} < \delta \leq 1. \tag{12}$$

*(For example, this is guaranteed if $p_{\text{train}}$ and $p_{\text{env}}$ arise via high-reward filtering as in Lemma 1.)*

*Consider deterministic policies $\pi : \mathcal{X} \to \{0, 1\}$ and define:*

- *The **SFT (imitation) objective***

$$\mathcal{L}_{\text{SFT}}(\pi) := \mathbb{E}_{x \sim p_{\text{train}}} \big[ \mathbf{1}\{\pi(x) \neq y(x)\} \big].$$

- *The **RL objective** (expected reward in the environment) with reward $r(x, a) = \mathbf{1}\{a = y(x)\}$:*

$$J_{\text{RL}}(\pi) := \mathbb{E}_{x \sim p_{\text{env}}} \big[ r(x, \pi(x)) \big] = \mathbb{E}_{x \sim p_{\text{env}}} \big[ \mathbf{1}\{\pi(x) = y(x)\} \big].$$

*Define the* trivial *and* robust *policies by*

$$\pi_{\text{triv}}(x) := \phi_{\text{triv}}(x), \qquad \pi_{\text{rob}}(x) := \phi_{\text{rob}}(x) = y(x).$$

*Then:*

- *(a) The SFT objective distinguishes the two policies only up to the small margin $\varepsilon_{\text{train}}$:*

$$\mathcal{L}_{\text{SFT}}(\pi_{\text{rob}}) = 0, \qquad \mathcal{L}_{\text{SFT}}(\pi_{\text{triv}}) = \varepsilon_{\text{train}}.$$

- *(b) The RL objective distinguishes them by the larger gap $\delta$:*

$$J_{\text{RL}}(\pi_{\text{rob}}) = 1, \qquad J_{\text{RL}}(\pi_{\text{triv}}) = 1 - \delta.$$

  *In particular, the improvement in SFT loss when going from $\pi_{\text{triv}}$ to $\pi_{\text{rob}}$ is $\varepsilon_{\text{train}}$, whereas the improvement in RL reward is $\delta > \varepsilon_{\text{train}}$.*

- *(c) For any policy $\pi$, we have*

$$J_{\text{RL}}(\pi) \leq 1 = J_{\text{RL}}(\pi_{\text{rob}}),$$

  *with equality if and only if $\pi(x) = y(x)$ for $p_{\text{env}}$-almost every $x$. Thus $\pi_{\text{rob}}$ is the unique optimal policy for the RL objective (up to $p_{\text{env}}$-null sets), while $\pi_{\text{triv}}$ is strictly suboptimal whenever $\delta > 0$.*

**Remark 2** (Interpretation for SFT vs. RL and distillation). *In the regime where the SFT teacher or data-generating process is strong but imperfect, it is natural to have $\varepsilon_{\text{train}}$ very small (rare disagreements between trivial and robust features on the observed SFT data), while the environment disagreement $\delta$ is larger, because harder or more diverse test prompts break the trivial shortcut more frequently.*

*One concrete interpretation is:*

- *The* trivial *feature $\phi_{\text{triv}}$ encodes the behavior of a teacher policy $\pi_T$, obtained for example by supervised finetuning or heuristics, with*

$$\pi_T(x) := \phi_{\text{triv}}(x), \qquad \Pr_{x \sim p_{\text{env}}} [\pi_T(x) = y(x)] = 1 - \delta \quad \text{with } 0 < \delta \ll 1.$$

- *The SFT dataset is produced by distilling this teacher: sample $x \sim p_{\text{train}}$ and label it with $\pi_T(x)$. Under high-reward filtering (Lemma 1), the teacher almost never disagrees with $y(x)$ on $p_{\text{train}}$, so $\varepsilon_{\text{train}}$ is small.*

*Proposition 1 then says:*

- *From the SFT objective's perspective, moving from $\pi_{\text{triv}}$ (the shortcut / teacher policy) to $\pi_{\text{rob}}$ only improves the loss by $\varepsilon_{\text{train}} \ll 1$, which can be negligible relative to optimization noise, model capacity, or inductive bias. This makes it easy for SFT to "overfit" to the trivial pattern.*

- *From the RL objective's perspective, the same move improves the reward by $\delta$, which can be substantially larger. Thus RL has a much stronger structural incentive to abandon the trivial shortcut and adopt the robust strategy $\pi_{\text{rob}}$, which is in fact the unique policy that maximizes expected reward.*

*The exact case $\varepsilon_{\text{train}} = 0$ corresponds to an idealized limit in which trivial and robust features agree on all SFT training examples; the general case $\varepsilon_{\text{train}} < \delta$ is what one expects in realistic pipelines with high-reward filtering and broader RL exploration.*

## K.1 PROOF OF LEMMA 1

*Proof.* Write

$$\alpha := \mu(H) \in (0,1), \qquad 1 - \alpha = \mu(L).$$

By the law of total probability,

$$\begin{aligned}
\delta &= \Pr_\mu[D] \\
&= \Pr_\mu[D \mid H]\,\mu(H) \,+\, \Pr_\mu[D \mid L]\,\mu(L) \\
&= \varepsilon_{\text{train}}\,\alpha \,+\, \Pr_\mu[D \mid L]\,(1 - \alpha).
\end{aligned}$$

We want to compare $\delta$ with $\varepsilon_{\text{train}}$. Subtracting, we obtain

$$\begin{aligned}
\delta - \varepsilon_{\text{train}} &= \varepsilon_{\text{train}}\,\alpha \,+\, \Pr_\mu[D \mid L]\,(1 - \alpha) \,-\, \varepsilon_{\text{train}} \\
&= (1 - \alpha)\,\Pr_\mu[D \mid L] \,-\, (1 - \alpha)\,\varepsilon_{\text{train}} \\
&= (1 - \alpha)\,\big(\Pr_\mu[D \mid L] - \varepsilon_{\text{train}}\big).
\end{aligned}$$

By assumption, $0 < 1 - \alpha < 1$ and

$$\Pr_\mu[D \mid L] \,>\, \Pr_\mu[D \mid H] = \varepsilon_{\text{train}},$$

so the product $(1 - \alpha)\big(\Pr_\mu[D \mid L] - \varepsilon_{\text{train}}\big)$ is strictly positive. Therefore

$$\delta - \varepsilon_{\text{train}} > 0 \quad \Longrightarrow \quad \varepsilon_{\text{train}} < \delta,$$

as claimed. $\qquad\square$

## K.2 PROOF OF PROPOSITION 1

*Proof.* We prove (a), (b), and (c) in turn.

**(a) SFT objective.** By definition of $y$ and $\pi_{\text{rob}}$, we have $\pi_{\text{rob}}(x) = y(x)$ for all $x \in \mathcal{X}$, so

$$\mathbf{1}\{\pi_{\text{rob}}(x) \neq y(x)\} = 0 \quad \text{for all } x.$$

Therefore

$$\mathcal{L}_{\text{SFT}}(\pi_{\text{rob}}) = \mathbb{E}_{x \sim p_{\text{train}}}\big[\mathbf{1}\{\pi_{\text{rob}}(x) \neq y(x)\}\big] = 0.$$

For $\pi_{\text{triv}}$, observe that

$$\pi_{\text{triv}}(x) \neq y(x) \quad \Longleftrightarrow \quad \phi_{\text{triv}}(x) \neq \phi_{\text{rob}}(x),$$

so

$$\mathcal{L}_{\text{SFT}}(\pi_{\text{triv}}) = \mathbb{E}_{x \sim p_{\text{train}}}\big[\mathbf{1}\{\phi_{\text{triv}}(x) \neq \phi_{\text{rob}}(x)\}\big] = \Pr_{x \sim p_{\text{train}}}\big[\phi_{\text{triv}}(x) \neq \phi_{\text{rob}}(x)\big] = \varepsilon_{\text{train}}.$$

This proves (a).

**(b) RL objective.** We similarly have

$$\mathbf{1}\{\pi_{\text{rob}}(x) = y(x)\} = 1 \quad \text{for all } x,$$

so

$$J_{\text{RL}}(\pi_{\text{rob}}) = \mathbb{E}_{x \sim p_{\text{env}}}\big[\mathbf{1}\{\pi_{\text{rob}}(x) = y(x)\}\big] = 1.$$

For $\pi_{\text{triv}}$, we use

$$\pi_{\text{triv}}(x) = y(x) \quad \Longleftrightarrow \quad \phi_{\text{triv}}(x) = \phi_{\text{rob}}(x),$$

to obtain

$$\begin{aligned}
J_{\text{RL}}(\pi_{\text{triv}}) &= \mathbb{E}_{x \sim p_{\text{env}}}\big[\mathbf{1}\{\phi_{\text{triv}}(x) = \phi_{\text{rob}}(x)\}\big] \\
&= 1 - \Pr_{x \sim p_{\text{env}}}\big[\phi_{\text{triv}}(x) \neq \phi_{\text{rob}}(x)\big] = 1 - \delta.
\end{aligned}$$

This proves (b).

**(c) Optimality of the robust policy for RL.** Let $\pi$ be any deterministic policy. For each $x \in \mathcal{X}$,

$$\mathbf{1}\{\pi(x) = y(x)\} \leq 1,$$

with equality if and only if $\pi(x) = y(x)$. Taking expectations under $p_{\text{env}}$ yields

$$J_{\text{RL}}(\pi) = \mathbb{E}_{x \sim p_{\text{env}}}\big[\mathbf{1}\{\pi(x) = y(x)\}\big] \leq \mathbb{E}_{x \sim p_{\text{env}}}[1] = 1 = J_{\text{RL}}(\pi_{\text{rob}}).$$

Hence $J_{\text{RL}}(\pi) \leq J_{\text{RL}}(\pi_{\text{rob}})$ for all $\pi$, so $\pi_{\text{rob}}$ is an optimal policy.

Moreover, if $J_{\text{RL}}(\pi) = 1$, then the above inequality must hold with equality, which is only possible if $\mathbf{1}\{\pi(x) = y(x)\} = 1$ for $p_{\text{env}}$-almost every $x$. Equivalently,

$$\pi(x) = y(x) \quad \text{for } p_{\text{env}}\text{-almost every } x.$$

Thus any maximizer coincides with $\pi_{\text{rob}}$ almost surely under $p_{\text{env}}$, and $\pi_{\text{rob}}$ is the unique optimal policy up to $p_{\text{env}}$-null sets.

Finally, since $\delta > 0$ by equation 12, we have

$$J_{\text{RL}}(\pi_{\text{triv}}) = 1 - \delta < 1 = J_{\text{RL}}(\pi_{\text{rob}}),$$

so $\pi_{\text{triv}}$ is strictly suboptimal. This proves (c) and completes the proof. $\qquad\square$

